# SMiR: A Synthetic Data Pipeline to Improve Multi-Image Reasoning

## Abstract

Vision-Language Models (VLMs) have demonstrated strong performance in single-image understanding, supported by many high-quality instruction datasets. However, multi-image reasoning tasks remain under-explored in the open-source community due to two major issues: (1) scaling up datasets with multiple correlated images and complex reasoning instructions is resource-intensive and difficult to maintain quality and (2) there is a shortage of robust multi-image evaluation benchmarks. To address these issues, we introduce SMiR, an efficient synthetic data-generation pipeline for multi-image reasoning, and a high-quality SMiR dataset generated using this pipeline. Our pipeline efficiently extracts highly correlated images using multimodal embeddings, combining visual and descriptive information and leverages open-source LLMs to generate quality instructions, offering a cost-effective alternative to expensive closed-source solutions. Additionally, we present SMiR-BENCH, a novel multi-image reasoning evaluation benchmark comprising 100 diverse examples across 7 complex multi-image reasoning tasks. Unlike existing benchmarks, SMiR-BENCH is multi-turn and allows for free-form responses, providing a more comprehensive evaluation of model expressiveness and reasoning capability. We demonstrate the effectiveness of SMiR dataset by fine-tuning several open-source VLMs and evaluating their performance on SMiR-BENCH. Our results show that models trained on our dataset outperform baseline models in multi-image reasoning tasks. Furthermore, we observe enhanced model expressiveness and more nuanced reasoning in free-form responses, highlighting the value of our approach for advancing open-source VLM research. [1]

## 1 Introduction

Vision-Language Models (VLMs) have shown impressive capabilities in tasks involving single images, particularly open-source models that have benefited from high-quality instruction datasets (Laurençon et al., 2024b; Zhang et al., 2023; Xu et al., 2022). However, when it comes to multi-image tasks, such as comparing or analyzing relationships between multiple images, the performance of open-source VLMs (Liu et al., 2024b;a; Li et al., 2024a; Awadalla et al., 2023; Yao et al., 2024; Wang et al., 2023b) lags significantly behind their closed-source counterparts in GPT-4 (Achiam et al., 2023), Claude 3.5 Sonnet (Anthropic, 2024a), Claude 3 (Anthropic, 2024b), and Gemini 1.5 (Reid et al., 2024). One of the crucial problems is that constructing large-scale, complicated multi-image reasoning datasets and evaluation benchmarks is challenging.

First, collecting and curating large-scale multiple images with high correlations is hard. Identifying correlated semantic information or entities across images requires large-scale images and sophisticated algorithms. Thus, most existing multi-image instruction tuning datasets do not have highly correlated images. For example, MANTIS (Jiang et al., 2024) often includes unrelated images within the same multi-image reasoning question, potentially undermining the complexity of the task. MMDU-45K (Liu et al., 2024e) attempts to address this by clustering image captions. MMInstruct (Liu et al., 2024d), on the other hand, only considers one image at a time, falling short

---

[1]Upon acceptance, we will open-source the synthetic data generation pipeline, our dataset, and evaluation benchmark.

of true multi-image reasoning. These shortcomings highlight the difficulties and needs for datasets featuring related images within multi-image scenarios.

Second, scaling up the number of highly correlated images presents significant challenges. Existing datasets such as MANTIS, MMDU-45K, and MultiInstruct require extensive human curation and annotation, resulting in a labor-intensive and time-consuming process. To minimize human effort, researchers have leveraged GPT-4 family models (Peng et al., 2023; Wang et al., 2023a) to generate synthetic datasets—including MMInstruct, Multimodal ArXiv (Li et al., 2024b), MIMIC-IT (Li et al., 2023a), StableLLaVA-Instruct (Li et al., 2023c), and SVIT-Instruct (Zhao et al., 2023). However, this method proves expensive and difficult to scale effectively.

Third, evaluating multi-image reasoning is complicated. Given the increased complexity of multi-image reasoning tasks, using multi-turn free-response evaluation instead of the multiple-choice format employed by previous benchmarks such as (Fu et al., 2024; Ying et al., 2024; Wang et al., 2024a; Yue et al., 2024; Singh et al., 2019; Hudson & Manning, 2019; Antol et al., 2015). Free-response evaluations are more challenging, requiring models to articulate their thought processes, providing insight into their reasoning abilities, and allowing for a more nuanced assessment of their capabilities.

To address these challenges, we propose a synthetic data generation pipeline, SMIR, for multi-image reasoning and a human-annotated evaluation benchmark for multi-image reasoning, SMIR-BENCH. SMIR aims to generate correlated and challenging multi-image reasoning questions, while SMIR-BENCH evaluates models on free-response, difficult multi-image scenarios.

To summarize, we address these issues with **our contributions**:

- Two novel sampling algorithms: Cluster Sampling for data quality robustness and Graph Iteration Sampling for diversity. Both use multimodal embeddings (combining image and caption) to group correlated images for challenging multi-image instruction tuning datasets.
- A scalable synthetic multimodal data generation pipeline utilizing open-source LLMs such as Meta Llama 3.1 70B Instruct Turbo (Dubey et al., 2024), eliminating the need for expensive closed-source models, reducing costs by up to 50 times (Kirkovska, 2024) and speeds up to by 10 times (Kirkovska, 2024), while significantly minimizing human annotation efforts.
- A new multimodal evaluation benchmark with free-form responses, assessing both final answers and reasoning processes in complex multi-image tasks. Using GPT-4-Turbo and other open-source models as reference, we see up to 11% improvement with the SMIR dataset.

## 2 RELATED WORKS

**Vision Language Models**   We focus on instruction tuning Vision-Language Models (VLMs) that utilize a pretrained Large Language Model (LLM) backbone because this approach is cost-effective and more accessible for the open-source community. Since the backbone responsible for language understanding is already trained, the overall training process becomes simpler and requires fewer resources. Our primary task involves aligning the vision encoder—typically architectures like Vision Transformer (ViT) (Dosovitskiy, 2020), SigLIP (Zhai et al., 2023), or CLIP (Radford et al., 2021)—with the LLM backbone. This alignment is facilitated through linear layers that connect the vision encoder to the backbone, enabling the integration of visual and textual information. For instance, BLIP-2 (Li et al., 2023b) uses OPT (Zhang et al., 2022) and FLAN-T5 (Chung et al., 2022) as backbones, MiniGPT-4 (Zhu et al., 2023) utilizes Vicuna (Chiang et al., 2023), and Qwen-2-VL (Wang et al., 2024b) employs Qwen-2-1.5B (Yang et al., 2024) as the language backbone. In this paper, we focus on creating a high-quality multi-image reasoning dataset for instruction tuning instead of large-scale interleaved pretraining datasets like OBELICS (Laurençon et al., 2024a), MINT-1T (Awadalla et al., 2024), and LAION-5B (Schuhmann et al., 2022).

**Multi-Image Reasoning Data**   Recent advancements in multi-image reasoning instruction tuning datasets include MANTIS (Jiang et al., 2024) and MMDU-45K (Liu et al., 2024e), both aiming to improve reasoning capabilities in VLMs. However, these datasets have limitations in their approaches. MANTIS randomly concatenates single image pairs from LLaVA-665k (Liu et al., 2024a),

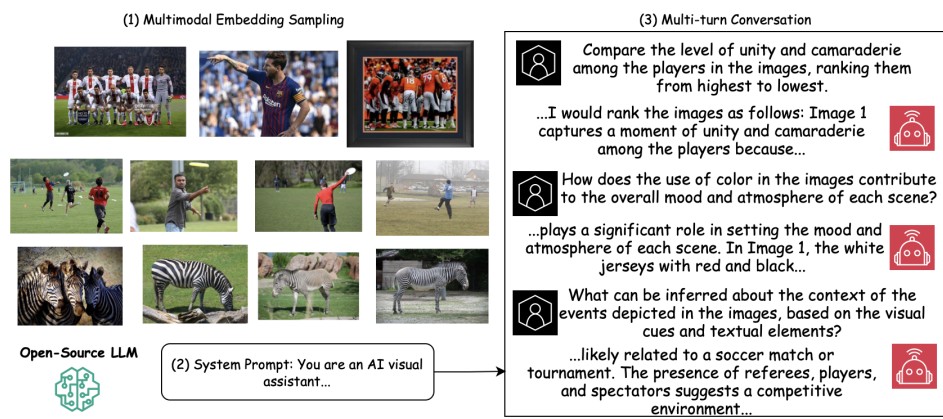

Figure 1: Our end-to-end pipeline: from sampling and LLM prompting to conversation generation. The example conversation is based on a sports scenario, demonstrating the pipeline's ability to generate contextually relevant multi-turn dialogues.

which often results in uncorrelated images within multi-image scenarios, potentially undermining the complexity of reasoning tasks. MMDU-45K attempts to address this issue by utilizing sentence transformers (Reimers, 2019) with description text and clustering techniques to group related images, but does not consider visual components. The dataset is then further enhanced, assisted by GPT-4 to generate comprehensive answers for the grouped images. Building upon these efforts, SMIR introduces a novel approach that leverages both vital visual and caption information to ensure highly correlated images within multi-image sets with the use of open-source LLMs. These scalable methods leads to the generation of more challenging questions that require deeper analysis and understanding of visual relationships, pushing the boundaries of multi-image reasoning capabilities in VLMs.

| Datasets | Multimodal Embedding | Correlated Images | Human-Annotation | Open-Source LLM |
|----------|----------------------|-------------------|------------------|-----------------|
| Mantis | No | No | Yes | No |
| MMDU | No | Yes | Yes | No |
| SMiR | Yes | Yes | No | Yes |

Table 1: Comparison of datasets highlighting key characteristics and methodologies.

**Multi-Image Reasoning Benchmarks**   Recent VLM benchmarks (Chiang et al., 2024; Lin et al., 2024; Liu et al., 2024c) have made strides by incorporating free-response evaluations, marking a significant improvement over traditional multiple-choice formats. However, these benchmarks still lack a comprehensive approach that combines automatic, multi-turn, and pairwise evaluation capabilities. Our benchmark addresses this gap, drawing inspiration from Auto-Hard-Auto v0.1 (Li et al., 2024c). We have adapted and expanded this framework to enable robust multimodal evaluation, providing a more holistic assessment of VLM performance across complex, multi-image reasoning tasks. This approach allows for a deeper analysis of both the final answers and the underlying reasoning processes employed by VLMs in real-world SMIR-BENCH scenarios.

## 3   SMIR: SYNTHETIC MULTI-IMAGE REASONING DATA PIPELINE

To generate complicated multi-image reasoning synthetic data efficiently, we introduce the SMIR pipeline. Given a large-scale of image-caption dataset $D$ with $N$ pairs of image-captions in $D$ as $(I_i, C_i)_{i=1}^N \in D$. SMIR constructs a multimodal embedding $E_i$ for each pair of $(I_i, C_i)$. Then, we apply grouping algorithms to find the correlations between multimodal pairs based on the embeddings. Finally, open-sourced LLMs are prompted to generate complicated question-answering

**SigLIP Embedding Space**  **CLIP Embedding Space**

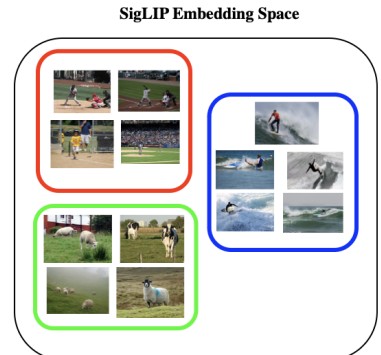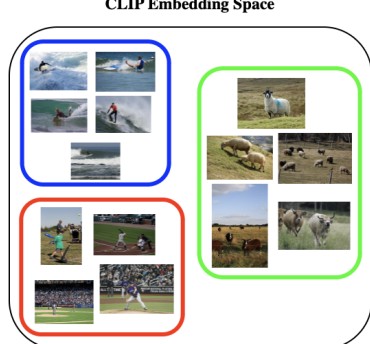

Figure 2: Cluster Matching from Different Embedding Spaces. Images are sampled only from the union of clusters with the same color.

pairs for on multiple pairs sampled based on correlations. In this section, we first introduce how we construct the multimodal embeddings, then we present the grouping algorithms, and finally we show how we generate the synthetic data samples via open-source LLM.

### 3.1 MULTIMODAL EMBEDDING CONSTRUCTION

To identify correlated images effectively, we developed a method that incorporates both visual and textual information from image-caption pairs. Our approach utilizes SigLIP and CLIP image embeddings alongside corresponding caption embeddings. We formulated a multimodal embedding by combining these components with a small constant, c, as follows:

$$E_{multimodal} = E_{image} + c \cdot E_{caption} \tag{1}$$

where $E_{multimodal}$ is the multimodal embedding, $E_{image}$ is the image embedding, and $E_{caption}$ is the caption embedding. For the ShareGPT4V (Chen et al., 2023), a $c = 0.2$ worked well, but this parameter may vary depending on the quality of individual image-caption pairs in other data sources.

Importantly, **relying solely on either image or caption information would limit our ability to concurrently consider both visual and textual contexts, which is crucial for establishing a comprehensive understanding of the images.** This multimodal approach enables us to capture the nuanced relationships between visual content and its associated descriptive text, thereby enhancing our capacity to identify and group correlated images effectively.

Following the generation of multimodal embeddings, we employed UMAP (McInnes et al., 2018) to reduce the dimensionality of the vectors. This technique allowed us to project the high-dimensional embeddings into a lower-dimensional space, facilitating more efficient analysis and visualization of the data while preserving its essential structure.

### 3.2 GROUPING IMAGES

We present two novel algorithms designed to group correlated images prior to leveraging an open-source Large Language Model (LLM) for synthetic data generation in multi-image reasoning tasks. The emphasis on correlated images is crucial, as it facilitates challenging multi-image reasoning scenarios. These scenarios require the model to identify intricate relationships and differentiate between visually similar scenes, thus enhancing the complexity and realism of the reasoning process.

**Clustering** We employed HDBSCAN (Malzer & Baum, 2020), a density-based clustering algorithm, to group the SigLIP and CLIP multimodal embeddings into coherent clusters. To establish meaningful relationships between the two embedding spaces, we developed a greedy algorithm that

Figure 3: Sampling based on embedding distance for increased diversity from initial image (orange). All images are considered within a single question.

matched SigLIP and CLIP clusters in a one-to-one fashion. This matching process ensured that each cluster from one embedding model corresponded to a semantically similar cluster from the other. The detailed steps of this Greedy Cluster Matching Algorithm in Figure 2 are presented in Algorithm 1 (Appendix A.1). Once clusters are matched, images are sampled from the cluster union until the desired number of images is selected.

**Vector Space Sampling**   We developed an iterative sampling method to select diverse yet related images within each embedding space. The process begins by randomly selecting an initial image, then iteratively sampling subsequent images based on their distance from previously selected points in the embedding space. This approach continues until the desired number of images is reached, ensuring a balance between diversity and semantic coherence in the selected image set. The detailed steps of this Random Sample Iteration algorithm in Figure 3 are presented in Algorithm 2 (Appendix A.2).

By assembling related images before prompting the LLM, we create a more coherent and contextually rich input, enabling the model to generate more nuanced and relevant synthetic data for multi-image reasoning tasks.

### 3.3 GENERATE SYNTHETIC DATA

Once grouped image-caption pairs are sampled, we take the caption embeddings and incorporate them into a system prompt for an open-source LLM, such as Meta Llama 3.1 70B Turbo, up to 50 times cheaper and 10 times faster compared to GPT-4 (Kirkovska, 2024). This process generates complex multi-turn conversations between User and Assistant as seen in Table 2 and questions tailored to the selected images, as shown in Figure 6 and Figure 7 (Appendix B).

## 4 DISCUSSION

Our approach involves several design choices, each with its own trade-offs. In this section, we discuss the decisions behind sampling algorithms, prompts, and data sources.

### 4.1 SAMPLING ALGORITHMS

Cluster-based algorithms demonstrate high efficacy in producing quality image-caption pairs, leveraging both SIGLIP and CLIP embeddings to confirm spatial relationships and associated semantic meanings. This dual-embedding validation ensures robust data quality, as images matched within

Table 2: SMIR Dataset Statistics

| Metric | Value |
|---|---:|
| Number of Samples | 160,000 |
| Maximum Number of Turns | 24 |
| Minimum Number of Turns | 2 |
| Average Number of Turns | 9.65 |
| Average Number of Images | 4.65 |
| Average User Tokens | 25.51 |
| Average Assistant Tokens | 124.32 |
| Open-Source LLM | Meta Llama 3.1 70B Turbo |

clusters are corroborated by two independent embedding models. However, this approach has a notable limitation: it can lead to overly specialized image subjects. This specialization occurs because sampling is confined to a single matched cluster, which restricts the diversity of selected images by excluding images from other clusters. A detailed example is presented in Figure 6 (Appendix B.1).

Vector sampling emerged as the preferred method due to its capacity to generate more generalized image subjects and foster diverse question generation when coupled with a system prompt. This approach allows for a wider range of image combinations, transcending the boundaries of individual clusters. Consequently, it facilitates the creation of more varied and cognitively demanding reasoning tasks. The flexibility of vector sampling in drawing from a broader semantic space contributes to a richer, more diverse dataset, potentially enhancing the complexity and applicability of subsequent machine learning tasks. A detailed example is presented in Figure 7 (Appendix B.2).

### 4.2 PROMPTS

In our data generation approach, we focused on creating two distinct types of questions: shorter, quick visual questions often involving OCR tasks, and longer reasoning questions that require in-depth analysis. Drawing inspiration from CoT (Wei et al., 2022), we designed prompts to generate multi-turn conversations, enhancing the complexity and depth of interactions. This dual approach necessitated the development of separate, tailored prompts for each question type, allowing us to effectively capture both complex reasoning scenarios and straightforward visual comprehension tasks. More details about the short prompt in Figure 8 (Appendix C.1 and long prompts Figure 9 (Appendix C.2).

### 4.3 DATA SOURCE

Our study leveraged the ShareGPT4V (Chen et al., 2023) dataset as the primary source for generating synthetic examples. This comprehensive dataset comprises of better image-caption pairs derived from LLaVA-Instruct (Liu et al., 2024a) and COCO (Lin et al., 2014). To maintain diversity from the data source, we synthetically generated 5,000 data points from each 20,000-image batch, resulting in a total of 160,000 synthetic examples. SMIR pipeline can also be applied easily to other data resources in the future.

## 5 MULTI-IMAGE BENCHMARK

SMIR-BENCH extends the Auto-Hard-Auto v0.1 (Li et al., 2024c) framework to the multimodal domain. It employs a judge model for pairwise comparison against a baseline model, evaluating responses on helpfulness, relevance, and conciseness. This approach enables a multi-turn, automatic, and challenging evaluation process.

### 5.1 MOTIVATIONS

We were motivated to create questions that more challenge VLMs to reason over multiple related images, analyze relationships, and derive meaning from series of images. We developed a multi-turn benchmark of 100 examples across seven diverse topics. This benchmark challenges models

to analyze relationships, derive meaning from image series, and provide hard explanations for complex visual tasks. Curated collaboratively by a human annotator and GPT-4, it uses images from the internet and Shot2Story (Han et al., 2023), compelling VLMs to demonstrate advanced visual reasoning capabilities beyond answering multiple choice.

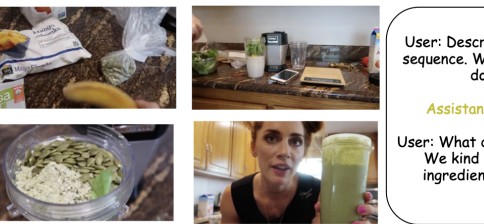

Figure 4: Evaluation Benchmark Using GPT-4o as Judge

## 5.2 EXPERIMENTS

We fine-tuned Mantis-8B-siglip-llama3-pretrained and idefics-8b on the SMIR dataset, using substantially less data (160K samples compared to the original 721K). These fine-tuned models were then evaluated on SMIR-BENCH. Remarkably, they outperformed both Mantis-8B-siglip-llama3 and Mantis-8B-Idefics2, despite the reduced training data. This improvement was evident when benchmarking against the closed-source GPT-4-Turbo, as well as against Mantis-8B-siglip-llama3 and Mantis-8B-Idefics2 themselves. These results demonstrate the effectiveness of our fine-tuning approach on the SMIR dataset, achieving superior performance with performance gains.

Table 3: Model Scores with GPT-4-Turbo Baseline

| Model Name | Score | Δ | 95% CI | Average Tokens |
|---|---|---|---|---|
| GPT-4o | 68.1 | | (-5.3, 6.4) | 442 |
| Claude-3.5-Sonnet-20240620 | 54.7 | | (-4.6, 7.0) | 374 |
| GPT-4-Turbo | 50.0 | | (0.0, 0.0) | 377 |
| Gemini-1.5-Pro | 38.7 | | (-7.0, 5.6) | 349 |
| Claude-3-Opus-20240229 | 31.0 | | (-5.7, 7.6) | 338 |
| **Mantis-8B-siglip-llama3-pretrained + SMiR-160k** | 9.5 | +3.4% | (-3.1, 3.1) | 180 |
| Mantis-8B-siglip-llama3 | 6.1 | | (-2.5, 2.6) | 170 |
| **Idefics2-8B + SMiR-50k** | 6.0 | +.6% | (-2.3, 2.9) | 173 |
| Mantis-8B-Idefics2 | 5.4 | | (-2.4, 2.3) | 195 |
| Idefics2-8B | 4.6 | | (-2.7, 2.0) | 122 |
| LLaVA-v1.6-mistral-7b-hf | 2.5 | | (-1.2, 1.6) | 361 |
| Mantis-8B-siglip-llama3-pretrained | 2.2 | | (-1.6, 1.8) | 198 |

Table 4: Model Scores with Mantis-8B-siglip-llama3 baseline

| Model Name | Score | Δ | 95% CI | Average Tokens |
|---|---|---|---|---|
| Claude-3-Opus-20240229 | 96.9 | | (-1.9, 2.0) | 338 |
| Claude-3-5-Sonnet-20240620 | 96.3 | | (-2.3, 1.4) | 374 |
| GPT-4-Turbo | 95.0 | | (-2.1, 2.5) | 377 |
| Gemini-1.5-Pro | 94.2 | | (-2.7, 2.1) | 349 |
| GPT-4o | 91.8 | | (-3.4, 3.1) | 442 |
| **Mantis-8B-siglip-llama3-pretrained + SMiR-160k** | 57.0 | +7.0% | (-7.9, 6.9) | 180 |
| Mantis-8B-siglip-llama3 | 50.0 | | (0.0, 0.0) | 170 |
| LLaVA-v1.6-Mistral-7B-HF | 18.9 | | (-4.0, 4.4) | 361 |
| Mantis-8B-siglip-llama3-pretraind | 11.7 | | (-4.2, 5.2) | 198 |

Table 5: Model Scores with Mantis-8B-Idefics2 Baseline

| Model Name | Score | Δ | 95% CI | Average Tokens |
|---|---|---|---|---|
| Claude-3-Opus-20240229 | 97.6 | | (-2.0, 1.4) | 338 |
| Gemini-1.5-Pro | 97.2 | | (-2.2, 1.4) | 349 |
| Claude-3-5-Sonnet-20240620 | 96.8 | | (-1.7, 1.5) | 374 |
| GPT-4-Turbo | 94.3 | | (-2.6, 2.4) | 377 |
| GPT-4o | 93.0 | | (-3.2, 2.8) | 442 |
| **Idefics2-8B + SMiR-50k** | 61.0 | +11.0% | (-7.4, 7.7) | 173 |
| Mantis-8B-Idefics2 | 50.0 | | (0.0, 0.0) | 195 |
| Idefics2-8B | 31.2 | | (-5.5, 5.9) | 122 |
| LLaAV-v1.6-mistral-7b-hf | 20.1 | | (-5.0, 3.9) | 361 |

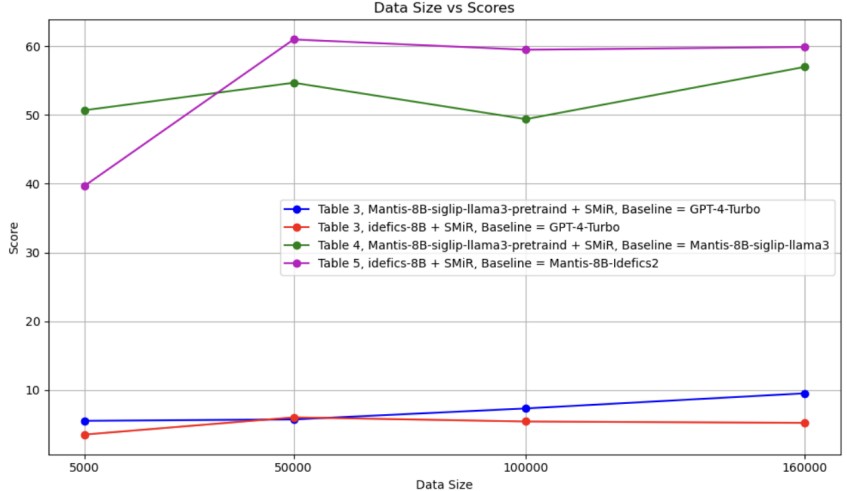

Figure 5: SMiR Dataset Size vs. Benchmark Score

## 6 CONCLUSION

This paper introduces a synthetic data pipeline designed to enhance multi-image reasoning capabilities on open-source VLMs. By leveraging multimodal embeddings and grouping algorithms, the pipeline generates high-quality synthetic multi-image reasoning instruction tuning data. The approach yields up to 11% improvement on SMiR-BENCH for popular open-source models, demonstrating the significant potential of synthetic data in advancing open-source VLM models.

**Limitations** Our methods have several limitations. Random sampling with iteration is time-intensive due to the need for recalculating distance embeddings for each new image sampled. Further investigation is needed to determine the scalability of our synthetic data. Future research should focus on developing more time-efficient algorithms and optimizing data mixtures.

**Broader Impact** This paper introduces a method for generating high-quality, cost-effective data for VLMs, addressing the growing challenge of data scarcity. By advancing these open-source techniques, we contribute to narrowing the performance gap between open and closed-source models, promoting more accessible and powerful multimodal AI.

## 7 REPRODUCIBILITY STATEMENT

The data sources are available on ShareGPT4V (Chen et al., 2023). Grouping algorithm codes can be found in (Appendix A, and prompts are provided in (Appendix C). All exact codes will be released and open-source.

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

# A ALGORITHM DETAILS

## A.1 GREEDY CLUSTER MATCHING ALGORITHM

We present the pseudocode for the Greedy Cluster Matching in Algorithm 1.

Let $C_S = \{S_1, ..., S_m\}$ and $C_C = \{C_1, ..., C_n\}$ be cluster sets from SigLIP and CLIP embeddings respectively. The algorithm proceeds as follows:

1. Select the largest cluster from either set: $X_{max} = \arg\max_{X \in C_S \cup C_C} |X|$

2. If $X_{max} \in C_S$, find the best match in $C_C$: $Y_{best} = \arg\max_{C_j \in C_C} score(X_{max}, C_j)$

3. If $X_{max} \in C_C$, find the best match in $C_S$: $Y_{best} = \arg\max_{S_i \in C_S} score(X_{max}, S_i)$

Where the score function is defined as:

$$score(A, B) = \frac{|A \cap B|}{\frac{|A|+|B|}{2}}$$

This process is repeated, greedily selecting the largest remaining cluster and finding its best match, until all clusters are matched or one set is exhausted.

---

**Algorithm 1** Greedy Cluster Matching Algorithm

---

**Require:** Two lists of clusters $c1$ and $c2$
**Ensure:** List of matched cluster pairs
 1: $c1 \leftarrow \text{sort}(c1, \text{key} = \text{len}, \text{reverse} = \text{True})$
 2: $c2 \leftarrow \text{sort}(c2, \text{key} = \text{len}, \text{reverse} = \text{True})$
 3: $\text{matched\_pairs} \leftarrow []$
 4: $\text{num\_samples} \leftarrow 0$
 5: **while** $c1$ is not empty and $c2$ is not empty **do**
 6:     **if** $\text{len}(c1[0]) \geq \text{len}(c2[0])$ **then**
 7:         $\text{larger\_cluster} \leftarrow c1.\text{pop}(0)$
 8:         $\text{smaller\_list} \leftarrow c2$
 9:     **else**
10:         $\text{larger\_cluster} \leftarrow c2.\text{pop}(0)$
11:         $\text{smaller\_list} \leftarrow c1$
12:     **end if**
13:     $\text{best\_match} \leftarrow \text{None}$
14:     $\text{best\_score} \leftarrow -1$
15:     **for** $i$, cluster in enumerate(smaller\_list) **do**
16:         $\text{overlap} \leftarrow \text{len}(\text{set}(\text{larger\_cluster}) \cap \text{set}(\text{cluster}))$
17:         $\text{avg\_size} \leftarrow (\text{len}(\text{larger\_cluster}) + \text{len}(\text{cluster}))/2$
18:         $\text{score} \leftarrow \text{overlap}/\text{avg\_size}$
19:         **if** score > best\_score **then**
20:             $\text{best\_score} \leftarrow \text{score}$
21:             $\text{best\_match} \leftarrow (i, \text{cluster})$
22:         **end if**
23:     **end for**
24:     **if** best\_match is not None **then**
25:         $\text{best\_index}, \text{best\_cluster} \leftarrow \text{best\_match}$
26:         $\text{union} \leftarrow \text{list}(\text{set}(\text{larger\_cluster}) \cup \text{set}(\text{best\_cluster}))$
27:         $\text{matched\_pairs}.\text{append}(\text{union})$
28:         $\text{num\_samples} \leftarrow \text{num\_samples} + \text{len}(\text{union})$
29:         $\text{smaller\_list}.\text{remove}(\text{best\_cluster})$
30:     **end if**
31: **end while**
32: **return** matched\_pairs, num\_samples

---

## A.2 RANDOM SAMPLING WITH ITERATION

We present the pseudocode for the Random Sampling with Iteration in Algorithm 2.

Let $X = \{x_1, ..., x_n\}$ be the set of embeddings.

$k$ is a parameter that determines the power of the distance calculation (default to 12), and $K$ is the desired number of selected embeddings.

1. Randomly select an initial embedding: $s_1 \in X$

2. Initialize selected set $S = \{s_1\}$

3. For $k = 2$ to $K$: $s_k = \arg\max_{x_j \in X \setminus S} \sum_{u \in S} \|x_j - x_u\|^k$ $S = S \cup \{s_k\}$

4. Return $S$

This formulation captures the process of iteratively selecting embeddings based on their cumulative distance from all previously selected embeddings, raised to the power $k$.

---

**Algorithm 2** Random Sampling with Iteration

---

**Require:**
1: $X$: Set of embeddings
2: $num\_samples$: Number of samples to select
3: $k$: Power factor for distance calculation (default: 12)
**Ensure:** Set of selected indices
4: $selected \leftarrow []$
5: $n \leftarrow |X|$                ▷ Number of embeddings
6: **for** $i = 1$ to $num\_samples$ **do**
7:   $distances \leftarrow \text{zeros}(n)$
8:   **if** $selected$ is empty **then**
9:    $sampled\_index \leftarrow \text{random\_integer}(0, n-1)$
10:   **else**
11:    **for** $j = 0$ to $n - 1$ **do**
12:     **if** $j \in selected$ **then**
13:      $distances[j] \leftarrow \infty$
14:     **else**
15:      $distances[j] \leftarrow \sum_{u \in selected} \|\|X[j] - X[u]\|\|^k$
16:     **end if**
17:    **end for**
18:    $inverted\_distances \leftarrow \frac{1}{distances + \epsilon}$       ▷ $\epsilon$ is a small constant
19:    $distribution \leftarrow \frac{inverted\_distances}{\sum inverted\_distances}$
20:    $sampled\_index \leftarrow \text{random\_choice}(\text{range}(n), p = distribution)$
21:   **end if**
22:   $selected.\text{append}(sampled\_index)$
23: **end for**
24: **return** $selected$

---

# B DATA SAMPLES

For the sake of brevity, we have included only two of the numerous multi-turn interactions present in each data sample.

## B.1 GREEDY CLUSTER MATCHING ALGORITHM

Samples obtained through Greedy Cluster Matching typically feature similar subjects and shot compositions, but when paired with carefully crafted prompts, these similarities can be leveraged to generate more challenging and nuanced questions.

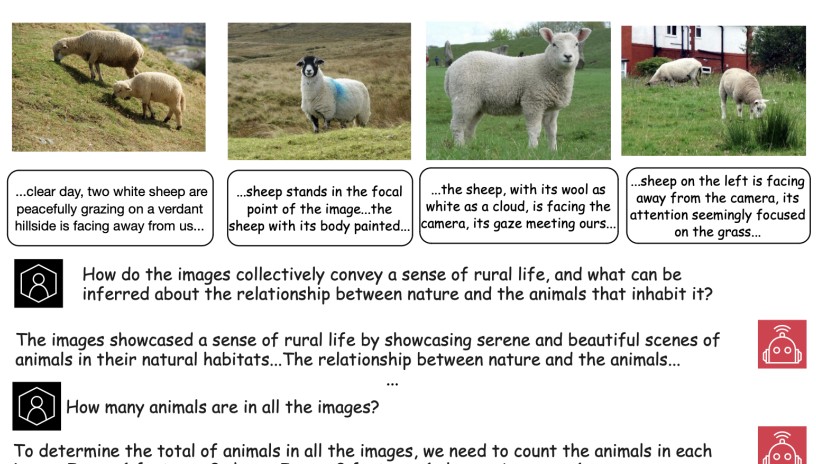

Figure 6: Images sampled from the same matched cluster often feature similar subjects or scenes.

## B.2 RANDOM SAMPLING WITH ITERATION

Random sampling tends to yield greater diversity in subjects compared to Greedy Clustering, striking a balance between variety and relatedness that can potentially lead to more robust and wide-ranging question sets.

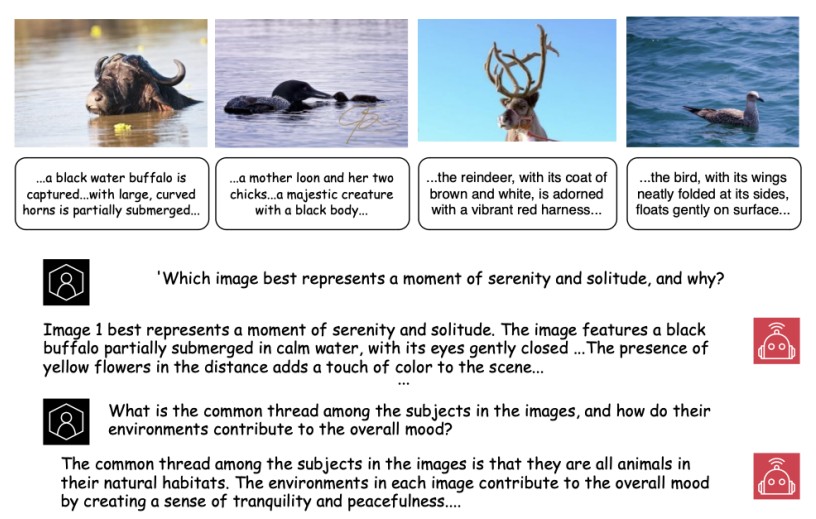

Figure 7: Images sampled using the iterative algorithm allow for different yet related subjects (e.g., various animal species)

## C  PROMPT

While prompts play a crucial role in data generation, optimizing them remains a significant challenge. After numerous iterations, we identified two particularly effective prompts for multi-image data generation.

### C.1  LLaVA STYLE PROMPT

Inspired by LLaVA (Liu et al., 2024b), our approach utilizes a specialized prompt to address simpler multi-image and single-image tasks, focusing on more straightforward visual comprehension and analysis.

> You are an AI visual assistant capable of analyzing multiple images, including both visual content and textual elements using Optical Character Recognition (OCR). You will receive four to five images, each potentially accompanied by captions and containing text, numbers, signs, or other recognizable characters. Your task is to create a plausible and challenging question that involves comparison, ranking, storytelling, logical reasoning, or detailed textual analysis across the images, and then provide a detailed answer...

Figure 8: LLaVA-style prompt for OCR and smaller visual task data generation

### C.2  LONGER PROMPT

Our approach aims to generate more complex, multi-turn questions that require in-depth reasoning across multiple images.

> Create questions that ask to compare elements across the images, such as identifying which image best represents a critical or turning point moment, quality, or characteristic; formulate questions that require ranking the images based on intricate and plausible criteria, such as strategic importance, sequence, or visual impact; develop questions that involve piecing together a narrative from the images, understanding a sophisticated sequence of events, or explaining a complex progression shown; and ask questions that require advanced logical reasoning to deduce why certain elements are present, the purpose behind actions shown, or the broader implications of what is depicted...

Figure 9: SMIR prompt to generate more complex question and answers

