# OpenReview forum: "SMiR: A Synthetic Data Pipeline To Improve Multi-Image Reasoning"
_ICLR.cc/2025/Conference — ICLR 2025 Conference Withdrawn Submission_

### Official Review · Reviewer_reGL · 2024-10-31

**Soundness:** 3
**Presentation:** 3
**Contribution:** 3
**Rating:** 5
**Confidence:** 4

**Summary:**

This paper introduces SMIR, a synthetic data pipeline designed to improve the performance of open-source visual language models (VLMs) on multi-image reasoning tasks by generating high-quality multi-image reasoning data.

Key Contributions: A data pipeline to generate high-quality multi-image reasoning questions and answers and a new benchmark with 100 diverse examples across seven complex multi-image reasoning tasks.

Data Pipeline: Multimodal Embedding: Utilizes SigLIP and CLIP to create multimodal embeddings of images and captions, effectively combining visual and descriptive information.Image Grouping Algorithms: Proposes two novel sampling algorithms—cluster-based sampling and graph-iteration-based sampling—to enhance data quality and diversity.

Benchmark: Unlike existing benchmarks, SMIR-BENCH is multi-round and allows for free-form responses, offering a more comprehensive evaluation of model expression and reasoning abilities.

**Strengths:**

The paper's strengths lie in its innovative approach to multi-image reasoning by using a synthetic data pipeline (SMIR) that enhances data quality and diversity. It introduces SMIR-BENCH, a comprehensive multi-image benchmark allowing for free-form responses, offering richer evaluation. Experimental results demonstrate significant performance improvements in open-source VLMs, highlighting SMIR's potential to advance open-source research in complex visual reasoning tasks.

**Weaknesses:**

The main drawbacks of this paper are its lack of emphasis on the inherent challenges and necessity of multi-image reasoning. Firstly, the pipeline seems predisposed to grouping similar images together, potentially leading to homogeneous questions like counting objects, which limits the diversity of reasoning questions that can be asked. Secondly, from an experimental perspective, the paper does not show whether this data confers additional capabilities to the model, such as improved performance on interleaved or video benchmarks following multi-image training.

**Questions:**

Based on the above discussion, I have two questions:

1. What is the difference between multi-image question answering and video question answering? In my view, video-based questions are even more challenging, as the model needs temporal awareness to answer correctly.

2. How does a model trained on this type of data perform on other different types of benchmarks, such as commonly tested image question answering (e.g., MMBench, MMStar) and interleaved image benchmarks?

---

### Official Review · Reviewer_Rscr · 2024-10-31

**Soundness:** 2
**Presentation:** 2
**Contribution:** 2
**Rating:** 3
**Confidence:** 5

**Summary:**

The author proposed a synthetic data scheme that can use open source models to synthesize a large amount of multi-image reasoning data. The author used this framework to build training sets and test sets and verified the effectiveness of the method.

**Strengths:**

The article proposes a method to synthesize multi-image reasoning data and verifies its effectiveness on the benchmark proposed by itself.

**Weaknesses:**

1. The layout of the article is rather confusing, and the main text is only eight pages long, which seriously affects the readability of the article

2. "... instead of the multiple-choice format employed by previous benchmarks ... " in Line 065 involves overclaim. MileBench[1], which is also a multi-image evaluation benchmark, also includes open-form questions.

3. "A scalable synthetic multimodal data ..." in Line 079 cannot be used as a contribution. Everyone knows that closed-source models have price issues, but they can be quickly accessed through the API, while open-source models also require GPU support. In addition, the author did not prove that this method can achieve similar quality as using closed-source models.

4. The research on Multi-Image Reasoning Benchmarks in Related work is incomplete, lacking MileBench[1], MMIU[2]

5. The author's experimental settings are wrong. If it is necessary to prove that the SMIR dataset generated by the author's proposed method can improve the model's multi-image understanding ability as a training set, it cannot be limited to the SMIR-BENCH experiment, which is not a zero-shot. Instead, more experimental results of other multi-image benchmarks are needed.


[1] MileBench: Benchmarking MLLMs in Long Context
[2] MMIU: Multimodal Multi-image Understanding for Evaluating Large Vision-Language Models

**Questions:**

please refer weaknesses for details.

---

### Official Review · Reviewer_mzUL · 2024-11-04

**Soundness:** 2
**Presentation:** 3
**Contribution:** 2
**Rating:** 5
**Confidence:** 5

**Summary:**

This paper proposes SMiR, a synthetic data pipeline to create multi-image training data. It first adopts a clustering strategy to group different multimodal semantics and uses a sampling strategy to construct multi-image pairs. After that, an open-source LLM is used to generate reasoning QA. Besides, a related benchmark with human annotation is also introduced. With this synthetic dataset, the paper shows some results to indicate the multi-image performance of finetuned VLMs.

**Strengths:**

1. The method is entirely synthetic and automatic, requiring no GPT API usage or human lobar. It's efficient and economical for scaling up training data for VLMs.

2. The idea is reasonable and practical. Using multimodal embeddings consider both image and text information for grouping. The adopted sampling strategy also works well.

3. The paper is clearly written and easy to follow.

**Weaknesses:**

1. The image domain is very limited. The paper only utilizes ShareGPT-4V to construct multi-image data, which mainly covers natural scenes and some text images. Instead, current VLMs are required to solve more diverse tasks and domains, such as table, plot, OCR-related, math, low-level and GUI. The author should give experiments to demonstrate the proposed approach is generalizable to most visual scenarios.

2. Since the capability of open-source LLM may not strong enough, the QA quality of constructed data cannot be ensured. The author should give human evaluation on a subset of generated data to ensure the reliability of data.

3. Some ablation study can be added. Why use SigLIP and CLIP? They two have similar contrastive pretraining target and may result in similar grouping results. How about combining the embeddings of SigLIP with DINOv2 or MAE that have different pretraining targets?

4. Mantis also introduces a new benchmark. It's better to provide performance comparison on that benchmark as well.

**Questions:**

See weakness

---

### Note · Authors · 2024-11-27

I have read and agree with the venue's withdrawal policy on behalf of myself and my co-authors.